# Are Iron Tailings Suitable for Constructing the Soil Profile Configuration of Reclaimed Farmland? A Soil Quality Evaluation Based on Chronosequences

**DOI:** 10.3390/ijerph19148235

**Published:** 2022-07-06

**Authors:** Wenjuan Jin, Han Wu, Zhongyi Wei, Chunlan Han, Zhenxing Bian, Xufeng Zhang

**Affiliations:** 1College of Land and Environment, Shenyang Agricultural University, Shenyang 110161, China; wjjin@stu.syau.edu.cn (W.J.); bsyogurt@163.com (H.W.); drweizy@163.com (Z.W.); zxf115@stu.syau.edu.cn (X.Z.); 2Key Laboratory of Trinity Protection and Monitoring of Cultivated Land, Shenyang 110161, China

**Keywords:** land reclamation, solid waste reuse, soil reconstruction, soil quality index (SQI), reclamation years

## Abstract

Iron tailings used as soil substitute materials to construct reclaimed farmland soil can effectively realize the large-scale resource utilization of iron tailings and reduce environmental risks. It is vital to understand the mechanisms affecting reclaimed soil quality and determine the appropriate pattern for reclamation with iron tailings. Thus, a soil quality index (SQI) was developed to evaluate the soil quality of reclaimed farmland with iron tailings in a semi-arid region. Soil samples were collected from two reclamation measures (20 cm subsoil + 20 cm iron tailings + 30 cm topsoil and 20 cm subsoil + 20 cm iron tailings + 50 cm topsoil) with reclamation years of 3 (R3), 5 (R5), and 10 (R10) at three soil depths (0–10, 10–20, and 20–30 cm) to measure 13 soil physicochemical properties in western Liaoning, China. Adjacent normal farmland (NF) acted as a reference. Results indicated that iron tailings were suitable for constructing the soil profile configuration of reclaimed farmland. SQI of reclaimed soil increased with the reclamation year, but it has not reached the NF level after 3 years, while it was better than NF after 5 years. The nutrient content of reclaimed soil increased with the reclamation year, but it still did not reach the NF level after 10 years. SQI of R10 (with 50 cm topsoil) was also better than NF but slightly lower than R5 (with 30 cm topsoil). For the semi-arid region with sticky soil texture, the topsoil thickness of reclamation was not the thicker the better, and 30 cm topsoil covered on iron tailings in western Liaoning could achieve a better reclamation effect than 50 cm.

## 1. Introduction

The exploitation of mineral resources supports rapid economic development while also causing substantial ecological and environmental problems, which has become one of the key challenges facing the sustainable development of the contemporary world [1,2,3]. Globally, surface mining destroys the regional farmland, forest, and landscape by creating huge overburden dumps and wastelands [4]. These overburdened materials mostly consist of large boulders, loose rock fragments and tailings, devoid of organic matter and nutrients, which are left unmanaged and create environmental pollution, commonly known as “mine spoil” [5,6]. Solid wastes such as mine spoil, if not properly treated and reused, would cause serious pollution to the environment and health risks [7,8]. Solid waste was a misplaced resource, and the level of recycling of resources is one of the important signs of social progress and a pathway toward green and sustainable development [9,10,11]. The report of the 19th National Congress of the Communist Party of China pointed out that we should ‘establish and improve the economic system for the development of a green low-carbon cycle’, and the circular economy is also called the 4R economy: Reduce, Reuse, Recycle, and Remanufacture [12]. Obviously, solid waste resource utilization is a typical connotation of a circular economy [3,13].

Iron tailings are a great quantity of solid waste discharged in iron ore mining and beneficiation, causing serious environmental problems such as land occupation, soil/water pollution and environmental damages, ecological risks, and economic losses [7,14]. Compared with developed countries, China started late in the reuse of solid waste resources such as iron tailings and has not yet formed a unique solid waste classification resource utilization mode [12]. Therefore, it is urgent to develop effective tailings utilization technologies [15]. Currently, comprehensive utilization pathways of iron tailings include the recovery of valuable components [16], production of concrete fillers and ceramsite, etc. [17]. Although these technologies could reuse iron tailings to a certain extent, issues such as land occupation, environmental pollution, and ecological risks still exist due to the low dosage. Large-scale solid waste resource utilization is the focus of green development and circular development [12].

There are lots of abandoned mining pits formed in China. Under the guidance of the concept of “Clear waters and green mountains are as good as mountains of gold and silver”, land reclamation and ecological restoration in mining areas have become an important work of ecological civilization construction. However, the lack of soil sources is the main problem faced by mining pit reclamation [18]. Filling reclamation engineering projects using mine spoil, such as iron tailings, can consume a lot of accumulated tailings, prevent the wastes from occupying land, and address the shortage of reclamation materials for mining wastelands [19]. Previous research has also shown that using large amounts of iron tailings to reclaim abandoned mining areas is a feasible way to dispose of waste iron tailings [19,20,21]. Especially in the western Liaoning of northern China, the iron ore mining has formed largely abandoned mining pits, and iron tailings, coupled with the regional barren soil layer, resulting in the lack of reclamation soil sources, which makes the use of iron tailings to fill and reclaim mining wastelands is a good way to realize land reclamation and the large-scale resource utilization of iron tailings [21]. Since 2010, iron tailings have been actively used to reclaim farmlands in western Liaoning, which has increased a lot of farmlands and consumed many accumulated iron tailings, and achieved good effects.

Once the iron tailings are physically reclaimed and utilized to support plant growth, they are considered “mine soils” [22]. Compared with natural soils, mine soils are pedogenically young and often characterized by poor soil structure, a lack of distinctive soil horizons, and nutrient-deprived conditions [23,24]. Therefore, the quality status of mine soil after reclamation should be focused on. In addition, farmland is a three-dimensional system [25]; the reasonable degree of soil profile configuration of reclaimed farmland plays a decisive role in the migration of soil water, fertilizer, gas, and heat and has significant effects on soil water infiltration, nutrient transfer, and solute transport, and will affect the material and energy cycle of the farmland system [26]. Soil reconstruction should pay attention to building a reasonable profile configuration while increasing farmland area to realize a healthy and sustainable farmland system. After more than ten years of reclamation practice, at present, the change of reclaimed soil quality after reclamation in western Liaoning is still unclear, and it is not clear which profile configuration is appropriate. Therefore, it is of great significance to analyze the soil quality of different reconstruction measures in different reclamation years to understand the reclamation status quo and improve the future reclamation process.

Soil quality evaluation can evaluate the progress and success of reclamation, which is one of the important indicators to appraise the quality of reclamation farmland [26]. However, inferring soil quality by merely measuring single or specific soil property is insufficient [27]. Accurate, repeatable, systematic, and transparent quantitative soil quality can enhance the interpretation and comparability between different reclamation years [28]. Among the many soil quality evaluation methods, Soil Quality Index (SQI) has become the most commonly used method because of its simple calculation and quantitative flexibility [29,30]. Although soil’s physical, chemical and biological properties can reflect soil quality, it is necessary to select the most representative indicators according to the research objectives, considering the factors such as cost and difficulty of test methods. Minimum Dataset (MDS) can use the least indicators to monitor and reflect changes in soil quality caused by changes in soil management measures, which has been widely used to evaluate soil quality [31]. Therefore, our study intends to evaluate the soil quality of the farmland reclamation with iron tailings in western Liaoning by using the SQI based on the MDS.

This research contributes to confirming the feasibility of solid wastes such as iron tailings can be recycled for constructing the soil profile configuration of reclaimed farmland in the existing literature through a soil quality evaluation based on chronosequence and revealing the reconstruction mechanisms of farmland reclaimed with iron tailings and the optimal reconstruction profile configuration. The specific objectives of this study were to (1) develop an SQI evaluation process and analyze the reclaimed soil quality indicators characteristics at different profile configurations and reclamation years; (2) evaluate the SQI of reconstructed farmland based on the MDS and determine the changing mechanisms and key impact indicators of the reclaimed soil quality; and (3) explore the optimal soil profile configuration of reclaimed farmland in western Liaoning and enhance our understanding of the farmland reclamation process with iron tailings to guide the reclamation technology improvement and the management of soil after reclamation.

## 2. Materials and Methods

### 2.1. Study Area

Our study was conducted on reclaimed farmland with iron tailings at a surface iron mining (Jianping Shengde Rixin Mining Co., Ltd.) in Jianping County, Chaoyang City, western Liaoning Province, China, at 41°45′ N, 119°37′ E (Figure 1), which is a region rich in iron ore resources, and mining wastes, such as iron tailings and waste rocks, occupy large amounts of land. This region is characterized by a semi-arid monsoonal climate with a mean annual temperature of 7.6 °C, mean annual precipitation of 467 mm, and a mean annual effective evaporation of approximately 1853 mm. According to the soil classification system of China, the soil in this region belongs to Hapli-Ustic Argosols [32]. The original farmland has a thin tillage layer (about 15 cm), sticky soil texture, and poor soil moisture regimes, which leads to the farmland being mostly medium and low yield fields. Aridity is the primary limiting factor of regional farmland quality.

Mining companies must reclaim mining wasteland in accordance with Chinese regulations, and due to the lack of reclamation soil sources in western Liaoning, Jianping sheng Rixin Mining Co., Ltd. combined waste iron tailings and mining stripping soil to reclaim mining wastelands as farmland. In the early stage of reclamation, due to the lack of systematic and scientific theoretical guidance, the specific reclamation schemes and soil profile configuration are different in different periods. At the beginning of reclamation, according to the correlation standard and reclamation practices, it was considered that the thicker the reclamation soil was, the better the effect was. The subsoil with a thickness of 20 cm was filled in the lowest layer, the iron tailings with 20 cm were filled in the middle layer as the soil moisture retention layer, and the tillage soil stripping from mining was covered with 50 cm as the topsoil to form a reconstructed soil profile. In recent years, due to the lack of soil sources, combined with the characteristics of regional farmland and crop growth conditions, the topsoil thickness was changed to 30 cm. On the one hand, the main purpose of filling reclamation with iron tailings is to construct a water retention layer to solve the limitation of water shortage in agricultural development in semi-arid areas. On the other hand, the iron tailings and topsoil are mixed by rotary tillage to improve the regional sticky soil texture.

### 2.2. Soil Sampling and Analysis

Through field investigation, the soil profile configuration of two typical iron tailings reclamation farmland was (1) 20 cm subsoil + 20 cm iron tailings + 30 cm topsoil and (2) 20 cm subsoil + 20 cm iron tailings + 50 cm topsoil. Soil samples were collected at 3-year-old reclaimed soil (R3), 5-year-old reclaimed soil (R5), and 10-year-old reclaimed soil (R10), respectively, to analyze the soil quality changes of different profile configurations under different reclamation years (Figure 1). The reference soil samples were collected from the adjacent normal farmland (NF), which was not affected by mining (Figure 1). For each sampling year, following the diagonal sampling method, 5 points were chosen in the sampling area to collect composite samples from depths of 0–10 cm, 10–20 cm, and 20–30 cm by the diagonal sampling method, and they are distributed in the four corners and the middle of the sampling area [33]. Each sample consisted of several subsamples that were collected at three different points. The same sampling method was also applied in NF. The samples were air-dried, shredded, and sieved through a 2 mm sieve before performing the following chemical and physical analyses (Table 1).

### 2.3. Soil Quality Assessment Methods

#### 2.3.1. Minimum Dataset for Soil Quality Evaluation

According to the research results of Bünemann et al. [37], combined with the experience and the specific situation of the study area, the total dataset (TDS) of soil quality was established, including 13 indicators (Table 1). In our study, principal component analysis (PCA) combined with Norm value and Pearson correlation analysis was used to select the soil indicators that can best reflect the soil quality characteristics and have significant indigenous effects on the evaluation results from the TDS as the minimum dataset (MDS) [38]. Norm value is the length of the vector norm of the indicator in the multidimensional space composed of components; the longer the length, indicating that the greater the comprehensive load of the indicator in all principal components, the stronger the ability to explain comprehensive information [39]. The formula is as follows (Equation (1)):(1)$$N_{ik}=\sqrt{\sum_{i=1}^{k}\left(\mu_{ik}{}^{2}\cdot \lambda_{k}\right)}$$
where *N_ik_* is the comprehensive load of the indicator *i* on the first *k* principal components with eigenvalue ≥1; *μ_ik_* is the load of the indicator *i* on the principal component *k*; *λ_k_* is the eigenvalue of the principal component *k*.

The factors with high eigenvalues and soil variables with high factor loading were assumed to be indicators that can foremost represent farmland soil [40]; hence, the retained principal components are selected according to the eigenvalue >1, and the loading values of the indicator was within 10% of the maximum loading value [41,42]. If a single component contains more than one soil attribute, the multivariate correlation coefficient is used to determine whether the variable is redundant. For variables with significant correlation, a variable with a high Norm value was selected for soil quality evaluation, and the rest were excluded. If the highly weighted variables are not correlated, each variable can be used for soil quality evaluation [26].

#### 2.3.2. Evaluation Model of Soil Quality Index (SQI)

SQI is a comprehensive reflection of soil function by calculating the weight and score of each soil quality evaluation index. The greater the value, the better the soil quality [37]. According to the positive and negative correlation between each soil quality evaluation index and soil quality, the membership function between the evaluation index and soil quality was established, and the membership degree of the index was calculated by Equations (2) and (3) [43,44,45]. Then, the role of each factor is calculated by using the factor load in principal component analysis, and their weights are determined by Equation (4). Finally, the comprehensive evaluation index of soil quality is calculated by Equation (5) through the weighted comprehensive method and addition multiplication [46].
(2)$$S_{i}=\frac{x_{ij}-x_{i-\min}}{x_{i-\max}-x_{i-\min}}(\mathrm{positive}\;\mathrm{index})$$
(3)$$S_{i}=\frac{x_{i-\max}-x_{ij}}{x_{i-\max}-x_{i-\min}}(\mathrm{negative}\;\mathrm{index})$$
where *S_i_* is the standard value of soil variable, $x_{ij}$ is the measured value of soil quality index *i* in the year *j*, $x_{i-max}$ is the maximum value of index *i*, and $x_{i-min}$ is the minimum value of index *i*.
(4)$$w_{i}=\frac{C_{i}}{C}$$
where *w_i_* is the weight of the soil quality index *i*, $C_{i}$ is the common factor variance, *C* is the sum of the common factor variance.
(5)$$\mathrm{SQI}=\sum_{i=1}^{n}w_{i}\cdot S_{i}$$

### 2.4. Data Analysis

The SPSS (Statistical Program for the Social Sciences, release 25.0) was used to perform a correlated statistical analysis of the data. All variables follow the normal distributions (tested with the Kolmogorov–Smirnov test at the *p*-value of 0.05). One-way ANOVA (analysis of variance) was carried out to compare the means of soil characteristics of normal farmland and reclaimed soil chronosequence sites. Differences between individual means were tested using DMRT (Duncan’s multiple range test) at *p* < 0.05 significance level. PCA-loaded variables were subject to Pearson correlation analysis. Origin 2021 was used for drawing.

## 3. Results

### 3.1. Feature of Soil Quality Evaluation Indicators

Results of soil physical and chemical analyses indicated that the characteristics of reclaimed soil quality improved with the increase in reclamation years (Figure 2, Figure 3, Figure 4 and Figure 5). BD is a sensitive indicator of soil compaction, and the soil with low BD is loose, which is beneficial to water storage, and vice versa. BD of reclaimed farmland was higher than that of NF in different reclamation years. At the depths of 0–10 cm, 10–20 cm, and 20–30 cm, BD of R3 was significantly increased by 26.37%, 18.26%, and 20.13% (*p* < 0.05), R5 was significantly increased by 19.71%, 16.48%, and 17.13% (*p* < 0.05), and R10 was significantly increased by 7.86%, 11.33%, and 11.54% (*p* < 0.05), respectively, compared with NF (Figure 2). In addition, BD increased with soil depth. At different soil depths, the differences in R3 and R5 were not significant (*p* > 0.05), but in R10, the difference was significant (*p* < 0.05). With the increase in reclamation years, BD at all levels decreased significantly. SWC of NF was the highest, reclaimed farmland was significantly lower than that of NF, and the differences between different reclamation years were significant (*p* > 0.05). With the reclamation year increase, SWC showed a continuous growth trend in each soil depth. The SWC of R10 at 0–10 cm, 10–20 cm, and 20–30 cm increased by 57.53%, 21.37%, and 20.75%, respectively, compared with R3. There was no significant difference in SWC of different soil depths within the same reclamation year. 

Clay of NF was the highest in all sample plots, and clay of reclaimed farmland in R3, R5, and R10 was significantly different from that of NF (*p* < 0.05) (Figure 3). There was no significant difference in clay among different reclamation years (*p* > 0.05), and with the increase in reclamation year, clay increased slightly. However, there were significant differences among different soil depths in the same reclamation year (*p* < 0.05). The silt was the highest in NF and the lowest in R5, and it was significantly different among each sample plot (*p* < 0.05), while it has no significant difference among different soil depths in the same reclamation year (*p* > 0.05). The sand was the lowest in NF and the highest in R5, and it was significantly different among each sample plot (*p* < 0.05). It has no significant difference among different soil depths in the same reclamation year (*p* > 0.05).

There were significant differences in pH between different sample plots and different soil depths (*p* < 0.05) (Figure 4). pH decreased with the increase in reclamation year at 0–10 cm and 10–20 cm depths, from 7.62 and 7.61 in R3 to 7.41 and 7.55 in R10, respectively. OM of NF was the highest, which was greater than 10.0 g/kg at 0–10 cm and 10–20 cm depths. With the increase in the reclamation year, OM increased significantly (*p* < 0.05), but OM was still less than 8.0 g/kg. In the same reclamation year, OM decreased significantly with the increase in soil depth (*p* < 0.05).

TN of NF was significantly higher than the other three reclaimed farmlands (Figure 5). At 0–10 cm, TN of R3, R5, and R10 were decreased by 73.39%, 56.88%, and 46.79%, respectively, compared with NF. TN increased significantly with the reclamation year increase (*p* < 0.05), but there was no significant difference at different soil depths in the same reclamation year (*p* > 0.05). TP was highest in NF and lowest in R3; it was 0.37 g/kg and 0.27 g/kg at 0–10 cm, respectively. TK of each reclaimed farmland was higher than NF, and it showed a significant downward trend with the increase in the reclamation year. AN of NF was significantly higher than that of each reclaimed farmland (*p* < 0.05). At 0–10 cm, 10–20 cm, and 20–30 cm, AN was 66.03, 48.56 mg/kg, and 43.21 mg/kg, respectively, but in reclaimed farmland, it was all less than 40 mg/kg, while with the increase in reclamation year, AN showed a significant upward trend. AP of NF was significantly higher than that of reclaimed farmland, and there was no significant difference between different reclamation years, but it was significantly different at 0–10 cm from 10–20 cm and 20–30 cm (*p* < 0.05). AK of NF was significantly lower than that of reclaimed farmland, and it was the highest in R3, which was 138.75 mg/kg, 94.22 mg/kg, and 108.98 mg/kg at 0–10 cm, 10–20 cm, and 20–30 cm, respectively. With the increase in reclamation year, AK decreased significantly. In addition, AK among different soil depths in the same reclamation year was significantly different (*p* < 0.05). In general, nitrogen and phosphorus increased with the reclamation year but were all lower than NF; potassium was significantly higher than NF after reclamation but decreased with the reclamation year.

### 3.2. Construction of MDS for Soil Quality Evaluation

In the results of PCA, the eigenvalues of the first three components were greater than 1, and their cumulative contribution rate reached 90.96%, indicating that the minimum dataset can replace the whole dataset for soil quality evaluation (Table 2). The first principal component variance was 60.25%, in which TN had the maximum loading value. The loading values of TN, BD, OM, and AN were within 10% of the maximum loading value, while TN had a high correlation with the other three variables (Figure 6), respectively, 0.914, 0.993, and 0.975 (*p* < 0.01); therefore, only TN in PC-1 was selected as the MDS. The variance of PC-2 was 19.83%. pH had the maximum loading value, and TP, AP, and clay loading values were within 10% of the maximum loading value. According to Figure 6, the correlation coefficients of pH and TP, AP were respectively −0.85 (*p* < 0.01) and −0.60 (*p* < 0.05), while the correlation between pH and clay is very low, and the correlation between clay and TP, AP were all very low, while TP and AP had a high correlation with 0.74 (*p* < 0.01), according to the Norm value, AP and clay were selected in MDS. The variance of PC-3 was 10.88%, silt and sand were within 10% of the maximum loading value, while the correlation coefficient of silt and sand was very high at −0.98 (*p* < 0.01), sand was selected in the MDS depending on the Norm value. The MDS of the soil quality evaluation of farmland soil constructed using iron tailings in the semi-arid region consists of TN, AP, clay, and sand.

Empirically, the main purpose of using iron tailings as the matrix to fill reclaimed farmland in areas with sticky soil texture was to improve soil texture, and it has the downside of weak capacity in holding fertilizer. Clay and sand are basic elements used to reflect soil texture. TN and AP are basic elements used to maintain crop growth. Accordingly, TN, AP, clay, and sand are suitable and essential for evaluating the soil quality of reclaimed farmland with iron tailings.

### 3.3. Soil Quality Evaluation Based on MDS

When performing PCA again for the four selected SQI evaluation indicators in MDS, each PC explained a certain amount (%) of the variation in the dataset (Table 3). TN has the highest contribution value in SQI, with a commonality of 0.833, followed by sand, with a commonality of 0.824, and the commonality of clay and AP is 0.559 and 0.393, respectively. The weight of TN, AP, clay, and sand was 0.319, 0.151, 0.214, and 0.316, respectively.

We calculated the scores of each index and sum the weighted scores of each variable to obtain the SQI value of each sample plot in 0–10, 10–20 cm, and 20–30 cm depths (Figure 7). SQI was significantly higher (*p* < 0.05) at 0–10 cm (0.454–0.636) than 10–20 cm (0.383–0.528) and 20–30 cm (0.262–0.504) of each sample plot. SQI of NF was 0.542, 0.528, and 0.262, respectively, at 0–10 cm, 10–20 cm, and 20–30 cm. Among three kinds of reclamation farmlands in different years, the SQI of R5 was the highest at 0.636 at 0–10 cm, followed by R10 (0.597). SQI of R3 was both the lowest at 0–10 cm (0.454) and 10–20 cm (0.383); it decreased by 16.24% and 27.46%, respectively, compared with NF, but increased by 24.05% at 20–30 cm. SQI of R5 was significantly improved (*p* < 0.05). At 0–10 cm and 20–30 cm, it was significantly higher than that in the normal farmland, increasing by 17.34% and 92.37%, respectively, and restored to the normal farmland level at 10–20 cm. SQI at 0–10 cm and 20–30 cm of R10 was also significantly better than that of NF, increasing by 10.15% and 86.26%, respectively, but was slightly lower than that of R5. There were significant differences between R5 and R3 but no significant differences between R5 and R10.

### 3.4. Applicability Verification of Soil Quality Evaluation Method Based on MDS

Generally, the soil quality can be evaluated with high accuracy through the TDS of the soil quality evaluation indicators. However, due to the numerous indicators, experimental analysis is complicated and time-consuming. The indicator dataset can be simplified through a series of statistical analyses, but it will lead to a decrease in evaluation accuracy. Therefore, it is necessary to verify the applicability of MDS of the evaluation indicator in a specific region or a specific soil. The common factor variance of each indicator of TDS was obtained by PCA, and then, the weight of each indicator of TDS was obtained (Table 3). The above method was used to analyze the soil quality of TDS.

SQI based on the MDS (MDS-SQI) and SQI based on the TDS (TDS-SQI) were used for regression analysis to verify the accuracy of the comprehensive value of soil quality based on the MDS (Figure 8). MDS-SQI and TDS-SQI met the linear regression relationship (*p* < 0.01), and the correlation coefficient was 0.840. The regression equation is *y* = 0.840*x* + 0.147 (*n* = 36, *R*^2^ = 0.712, *p* < 0.01), where *y* represents TDS-SQI, *x* represents MDS-SQI. The above analysis shows that MDS can replace TDS, and the quality evaluation of farmland soil reclaimed by iron tailings through the MDS indicator system has high accuracy.

## 4. Discussion

### 4.1. Chronosequence Evolution of Soil Quality in Reclaimed Farmland

Soil is an interconnected system, and the reconstruction process contains a series of chain reactions that take a certain amount of time [1]. Soil texture was improved after reclamation with iron tailings. Compared with NF, the most significant features of the reclaimed soil in each reclamation year were high sand and low clay. Sand had the largest value in each reclaimed soil, and clay was significantly lower than NF (Figure 7). This was because the soil in the study area is cinnamon soil with high clay content. The iron tailings were filled under the topsoil and would be mixed into topsoil by tillage during the crop planting. Iron tailings were mostly irregular granular [47], and their specific surface area was larger than that of clay particles, increasing the sand content. Meanwhile, lots of iron tailings reduce the clay content and regulate the soil mechanical composition, which fully demonstrates that reclaiming farmland with iron tailings can effectively improve the soil texture in western Liaoning, and the results were consistent with those of Yang [48].

The reclaimed soil quality was poor in the early stage of reclamation; SQI of R3 was significantly lower than NF because TN and AP of R3 were significantly lower than NF (Figure 5). Soil nitrogen and phosphorus are essential elements for plant growth, and the phosphorus content affects soil fertility and physical and chemical characteristics such as SWC, pH, and OM [49]. Although TN of reclaimed soil showed an overall upward trend with the reclamation year, it was always smaller than NF. TN of R10 was only 53% of NF, and the nitrogen supply capacity was significantly poor. The variation trend of AP was consistent with TN and showed an upward trend with the reclamation year, but it was also less than NF. AP of R10 was only 70% of NF, and the phosphorus supply capacity was significantly poor, which was consistent with Li et al. [50], Duo and Hu [51], and Li et al. [52], that TN and AP showed an upward trend with the reclamation year, but it was always lower than that of normal farmland. The texture of iron tailings is sandy, and the water and fertilizer conservation abilities are poor. The nutrient of reclaimed soil was low, and the recovery time was long. The accumulation of soil nutrients is a long-term process, and it needs to be improved by changing the irrigation mode, rational planting, and following the principle of small amounts and multiple applications of fertilizer.

After 5 years of reclamation, SQI was significantly higher than NF, indicating that the reclamation with iron tailings has a great influence on the comprehensive quality of soil, and the reclaimed soil after 5 years can reach or even better than the comprehensive quality of normal farmland in the region. The research results of Mukhopadhyay et al. [6] also showed that the quality of reclaimed soil improved with the increase in reclamation year. Cao et al. [53] reported that the reclaimed soil was largely restored after 12 years of reconstruction; however, the recovery was not completed.

### 4.2. Effect of Profile Configuration on Reclaimed Soil Quality

Reclaimed soil quality will recover better with the increase in reclamation year [6], but the SQI of R10 was slightly less than R5. By comparing their profile configurations, R5 covered 30 cm topsoil on iron tailings, while R10 covered 50 cm topsoil. Through field investigation, local farmers plowed farmland with a depth of 30–40 cm by rotary tillage machine in spring. Therefore, the topsoil with 30 cm would mix iron tailings in the plowing process, which effectively increases the sand content of the topsoil and improves the sticky soil texture, and the number of iron tailings mixed in the topsoil increases with the years of cultivation and the soil texture was getting better and better (Figure 9). However, iron tailings were difficult to be mixed into the topsoil with 50 cm through plowing, so the effect of using iron tailings to improve the sticky soil texture was poor.

In addition, the sticky soil texture in the region leads to poor soil water retention capacity, and aridity is the primary limiting factor of regional farmland quality [48]. In the reclamation process, the role of filling iron tailings in the middle layer is to use the pore structure of iron tailings to build a water retention layer and improve the water holding capacity of the reclaimed soil. The research results of Jia [54] showed that in farmlands similar to the soil texture of our study area when the rainfall is less than 20 mm, the infiltration depth is 20 cm; when the rainfall is 20–50 mm, the infiltration depth is 40 cm, and when the rainfall is more than 50 mm, the infiltration depth can reach the soil layer below 40 cm. The mean annual precipitation in Jianping County is 467 mm, with the highest rainfall in July, about 137 mm. Generally, the maximum individual rainfall is less than 50 mm, so it is estimated that the infiltration depth of an individual rainfall in the study area will not exceed 40 cm. Therefore, the soil moisture migration might make it difficult to reach the iron tailings layer with 50 cm topsoil, which makes the soil water retention effect of the iron tailings layer poor, while the soil moisture could reach the iron tailings layer with 30 cm topsoil and effectively maintain the soil moisture, thereby promoting the nutrient cycling in soil (Figure 9). The amount of iron tailings mixed in the topsoil increases with the years of cultivation, which made the soil moisture retention ability stronger and the nutrient cycling ability better. As shown in Figure 2b, SWC after reclamation for 5 years was significantly higher than that after 3 years.

Our result is different from some previous cognition, that was, the thicker the reclaimed soil in land reclamation is, the better it is [55]. For semi-arid regions, the topsoil thickness when iron tailings are used for reclamation is not the thicker, the better, but the thickness suitable for regional conditions is the best. In western Liaoning, based on regional soil characteristics and tillage practice, when iron tailings are used as mine soil to reconstruct farmland, 30 cm topsoil covered on iron tailings can achieve a good reclamation effect, and with the increase in reclamation year, it is conducive to improving the quality of reclaimed soil. This reclamation pattern consumes large amounts of iron tailings and forms a soil water-retaining layer, which meets the requirements of regional crop cultivation and forms a good reclamation effect. Moreover, reducing the coverage thickness of the topsoil can effectively save soil resources and reclamation costs [56].

### 4.3. The Variation Regulation of Soil Quality in Vertical Profiles with Different Reclamation Years

The MDS-SQI was significantly higher (*p* < 0.05) at 0–10 cm (0.454–0.636) than 10–20 cm (0.383–0.514) and 20–30 cm (0.325–0.504) (Figure 10) of the reclaimed farmland. The MDS-SQI observed for NF was 0.542, 0.528, and 0.262, respectively, at 0–10 cm, 10–20 cm, and 20–30 cm. It showed that the quality of topsoil in reclaimed soil was higher than that of subsoil. In a similar study on reclaimed sites, the quality of topsoil was higher than that of subsoil [57]. For both the soil layers, the overall trend of SQI has followed the order: R3 < NF < R10 < R5. After 5 years of reclamation with iron tailings, the soil comprehensive quality at each soil depth could reach or even better than the level of regional normal farmland.

### 4.4. Key Indicator Identification and Policy Implications

By using a series of soil quality indicators, the effect of filling reclamation with iron tailings on soil quality was studied through the SQI method based on MDS, and the applicability of this method in the region was verified. The MDS was screened by PCA combined with the Norm value, and the Norm value was introduced to consider the load of the indicator on all principal components to avoid the loss of information on other principal components [58]. The summary of the relevant research results of soil quality evaluation based on MDS by Bünemann et al. [37] showed that bulk density, pH, organic matter, sand percentage, silt percentage, total nitrogen, available phosphorus, and soil water content have a high frequency of use. The three indicators (sand, TN, AP) determined in our study were consistent with most of the results. In addition, clay was selected as MDS in our study, indicating that in addition to sand, TN, and AP, the effect of clay content on soil quality in the study area was also significant, which was determined by the regional natural conditions and soil characteristics. The study area is a typical semi-arid region, the soil texture was sticky, and filling reclamation with iron tailings would change the soil texture. Therefore, the four MDS indicators selected in our study are the key indicators affecting the quality of reclaimed soil in western Liaoning.

The traditional filling reclamation materials such as fly ash and gangue usually contain harmful heavy metal elements such as Cd, Pb, and Hg, and the reclaimed soil has potential ecological harm [26]. In addition, if these heavy metals diffuse with runoff, they will also pollute a wider range of soil [51]. The results of our previous study found that the content of heavy metals such as Cd, Cr, Cu, Zn, Pb, Ni, Hg, and As did not exceed the risk intervention value for pollution of agricultural land in China [59], even lower than the heavy metal content in the regional native soil, with no toxicity and are not a source of pollution for the soil and crops [48]. In addition, our previous column leaching test results also showed that the heavy metal content in the leachate was very low, which met the standard [59], especially the content of Cr, Pb, and Cd was very low, which was almost impossible to be detected. This once again proves that it is feasible to use iron tailings as reclamation materials of mine wasteland, which can increase the farmland area and reduce the tailings accumulation, and save the cost of tailings pond management.

In general, our study evaluated the reclaimed soil quality after the implementation of different iron tailings reconstruction farmland techniques to identify the most effective reclamation techniques. The results of this study can also provide valuable policy implications on improving the treatment of waste iron tailings, guiding the formulation of land reclamation technical standards, and promoting ecological restoration of the mining area. For example, in the future formulation of relevant technical standards for land reclamation in China, the appropriate thickness of topsoil cover should be reasonably determined according to the actual situation of reclamation areas. In addition, the research results have a positive guiding role in the formulation of technical policies for “harmless”, “reduction”, and “resource utilization” of solid waste in China. Furthermore, through the wide promotion and application of this technology, the use of iron tailings to reclaim historical legacy mines in the region can effectively solve the problems of limited reclamation resources and shortage of repair funds faced by local governments in the ecological restoration of historical legacy mines and improve the comprehensive utilization value of abandoned lands. Our research results are of great significance to promoting the ecological restoration of mines, realizing the sustainable development of ecological civilization construction, and supporting the “UN decade on Ecosystem Restoration” action.

## 5. Conclusions

Iron tailings were confirmed to be suitable as soil substitutes for constructing the soil profile configuration of reclaimed farmland. The comprehensive quality of reclaimed soil improved with the reclamation year, but it has not reached the level of regional normal farmland after 3 years of reclamation. The soil quality after 5 years of reclamation was better than that of normal farmland. SQI of R10 was also better than NF but slightly lower than R5. The quality of topsoil was better than that of subsoil in the same reclaimed farmland. The thickness of topsoil would affect the reclaimed soil quality. The soil quality of 30 cm topsoil covered in 5 years of reclamation was better than that of 50 cm topsoil covered in 10 years of reclamation. For the semi-arid region with sticky soil texture, the thickness of reclaimed topsoil is not the thicker, the better. The topsoil covering 30 cm after iron tailings filling in western Liaoning could achieve a better reclamation effect; the topsoil texture was improved, and the reclamation cost was effectively saved.

Our study mainly analyzes the effect of the measures that have been completed by regional mines to reclaim farmland with iron tailings. The study was not carried out based on the strict and systematic experimental scheme design but conducted targeted research according to the actual reclamation process with iron tailings. According to the results of our study, it is expected to further establish a test site for systematic research in the study area in the future, to provide a basis for the improvement of ecological restoration theory and waste resource utilization technology in mining areas. 

## Figures and Tables

**Figure 1 ijerph-19-08235-f001:**
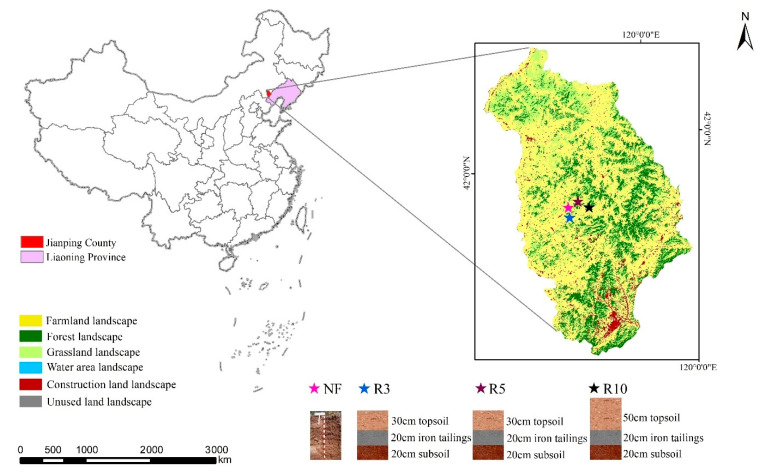
Location of the study area. Note: the color pentagrams represent the distribution of soil sample plots of reclaimed farmland. NF represents the adjacent normal farmland as a reference; R3 represents farmland reclaimed for 3 years with a soil profile configuration of 20 cm subsoil + 20 cm iron tailings + 30 cm topsoil; R5 represents farmland reclaimed for 5 years with a soil profile configuration of 20 cm subsoil + 20 cm iron tailings + 30 cm topsoil; R10 represents farmland reclaimed for 10 years with a soil profile configuration of 20 cm subsoil + 20 cm iron tailings + 50 cm topsoil. The photos of the map were adopted from www.google.com/maps (accessed on 27 May 2022).

**Figure 2 ijerph-19-08235-f002:**
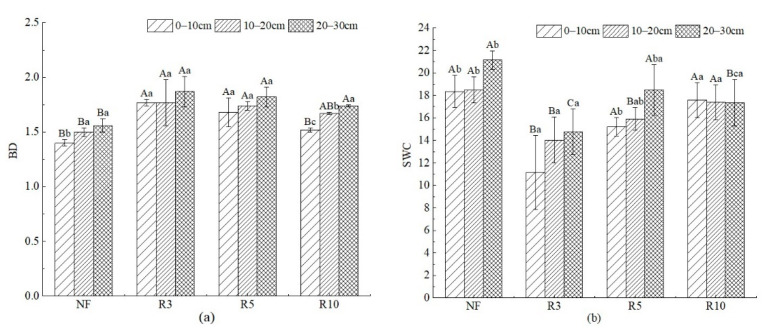
BD and SWC of soil in different depths with different reclamation years. Note: (**a**) BD; (**b**) SWC. Different capital letters indicate significant differences (*p* < 0.05) among different reclamation years in the same soil layer, and different lowercase letters indicate significant differences (*p* < 0.05) among different soil layers in the same reclamation year. See Table 1 for abbreviations and units.

**Figure 3 ijerph-19-08235-f003:**
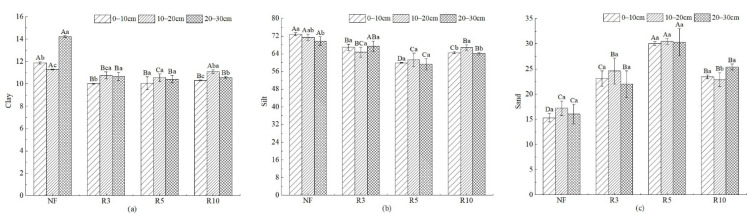
The mechanical composition of soil in different depths with different reclamation years. Note: (**a**) Clay; (**b**) Silt; (**c**) Sand. Different capital letters indicate significant differences (*p* < 0.05) among different reclamation years in the same soil layer, and different lowercase letters indicate significant differences (*p* < 0.05) among different soil layers in the same reclamation year. See Table 1 for abbreviations and units.

**Figure 4 ijerph-19-08235-f004:**
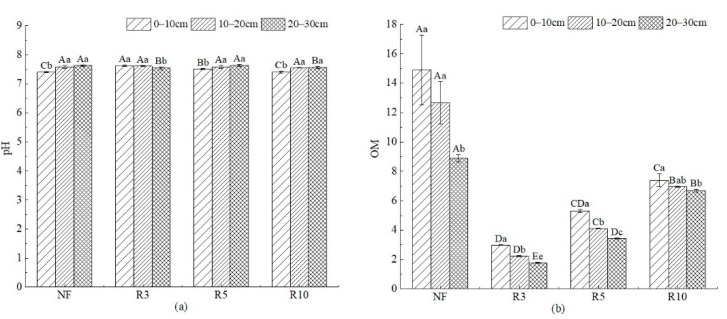
pH and OM of soil in different depths with different reclamation years. Note: (**a**) pH; (**b**) OM. Different capital letters indicate significant differences (*p* < 0.05) among different reclamation years in the same soil layer, and different lowercase letters indicate significant differences (*p* < 0.05) among different soil layers in the same reclamation year. See Table 1 for abbreviations and units.

**Figure 5 ijerph-19-08235-f005:**
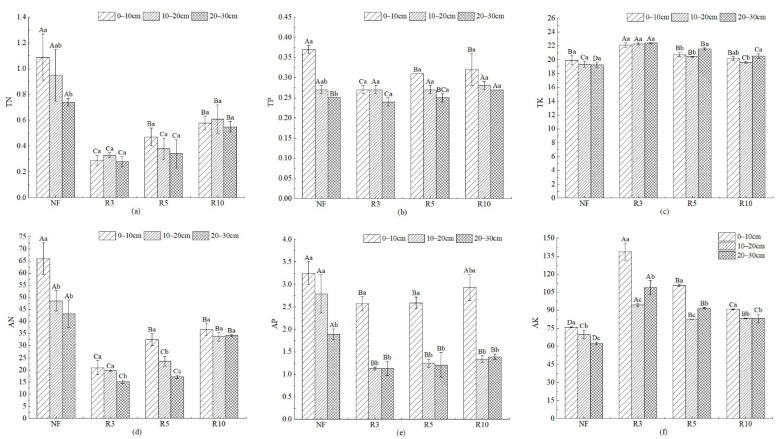
The soil nutrient in different depths with different reclamation years. Note: (**a**) TN; (**b**) TP; (**c**) TK; (**d**) AN; (**e**) AP; (**f**) AK. Different capital letters indicate significant differences (*p* < 0.05) among different reclamation years in the same soil layer, and different lowercase letters indicate significant differences (*p* < 0.05) among different soil layers in the same reclamation year. See Table 1 for abbreviations and units.

**Figure 6 ijerph-19-08235-f006:**
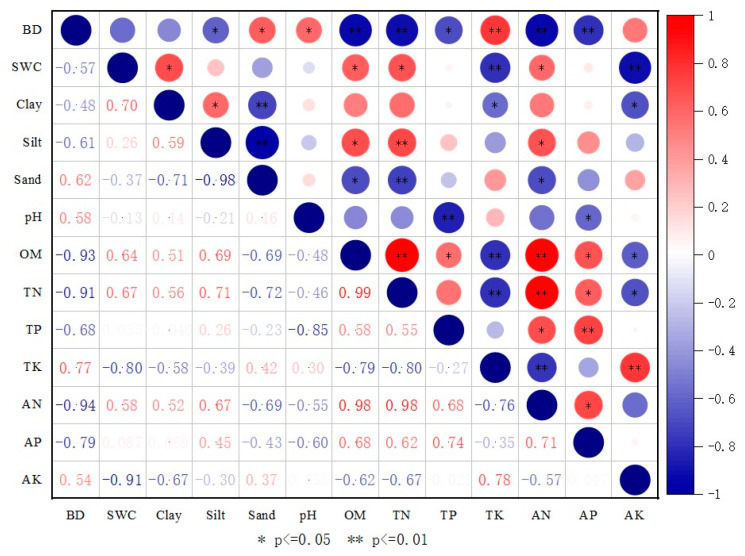
Correlations matrix for soil properties. Note: Color legend and circles illustrate the correlation coefficient values. Stars indicate the significance level of the correlation (*—correlation is significant at *p* < 0.05; **—correlation is extremely significant at *p* < 0.01).

**Figure 7 ijerph-19-08235-f007:**
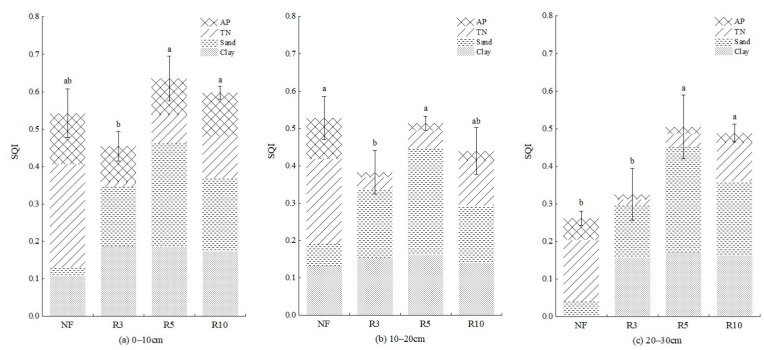
Contributions of individual soil indicator parameters to overall SQI under different sample plots. Note: (**a**) 0–10 cm depth; (**b**) 10–20 cm depth; (**c**) 20–30 cm depth (different letters indicate significant differences (*p* < 0.05) in the studied variables among different sample plots.

**Figure 8 ijerph-19-08235-f008:**
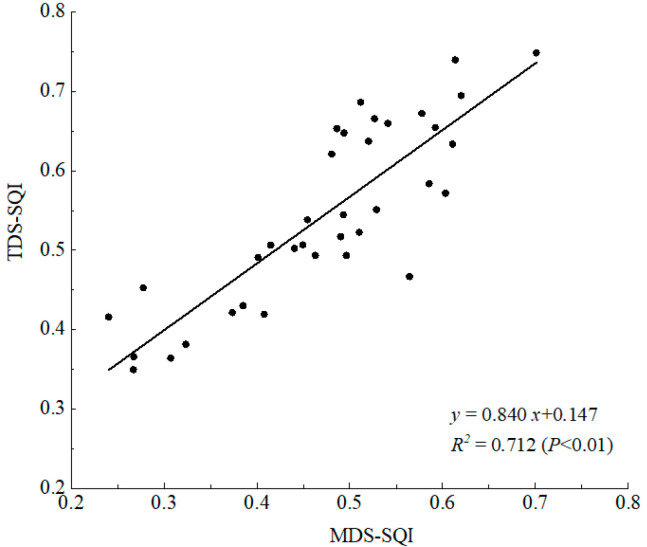
Relationship between MDS and TDS.

**Figure 9 ijerph-19-08235-f009:**
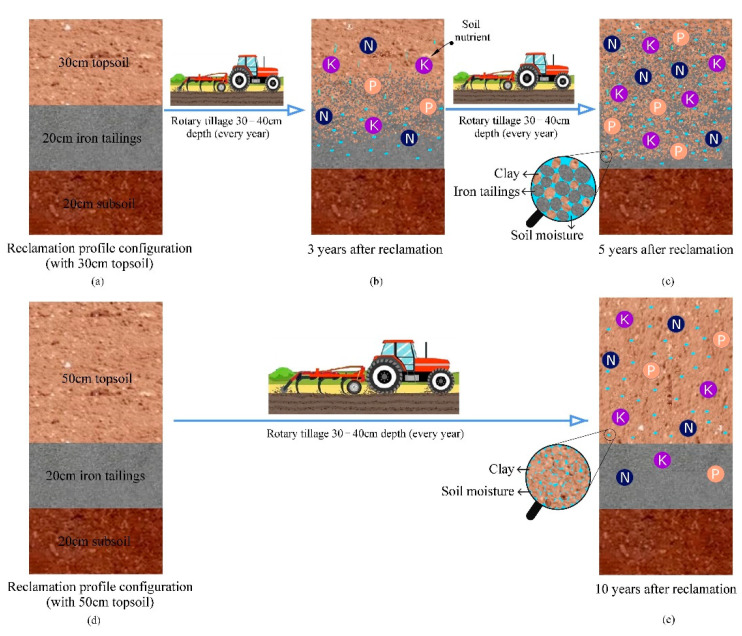
Schematic diagram of effects of different profile configurations on reclaimed soil quality. Notes: (**a**) reclamation soil profile configuration with 30 cm topsoil commonly used in recent years of reclamation; (**b**) soil changes after 3 years of reclamation with 30 cm topsoil, iron tailings mixed with topsoil through plowing 30–40 cm every spring, the clay content of topsoil decreased, soil porosity of topsoil increased and soil moisture storage capacity improved; (**c**) soil changes after 5 years of reclamation with 30 cm topsoil, more iron tailings was mixed into the topsoil with the increase in tillage years, and the soil moisture storage capacity was better; (**d**) reclamation soil profile configuration with 50 cm topsoil in the early stages of reclamation; (**e**) soil changes after 10 years of reclamation with 50 cm topsoil, iron tailings was difficult to be mixed into the topsoil through plowing 30–40 cm, but annual tillage could increase the porosity of topsoil and improve soil moisture storage capacity.

**Figure 10 ijerph-19-08235-f010:**
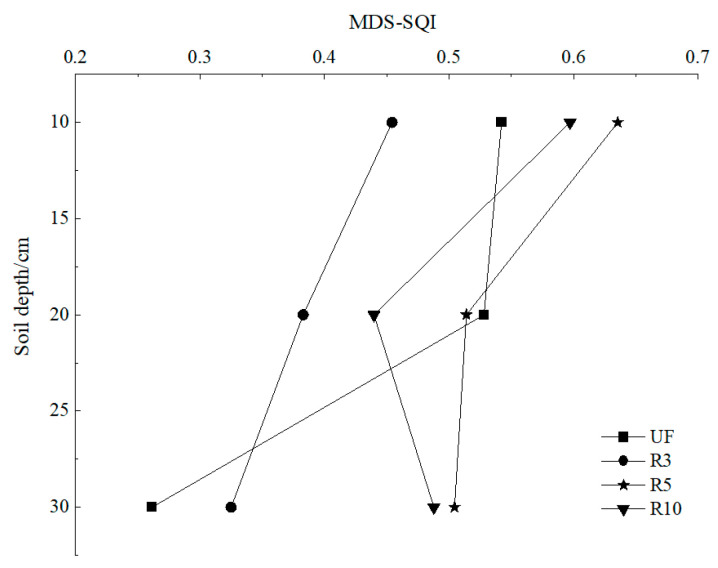
The trend of the profile of MDS-SQI for different years of reclamation.

**Table 1 ijerph-19-08235-t001:** Methods used to measure soil attributes for soil quality assessment.

Soil Properties	Abbreviation	Unit	Method
Bulk density	BD	g/cm^3^	Core cutter method [34]
Soil water content	SWC	%	Oven drying method [34]
Clay	Clay	%	Laster particle sizer method [34]
Silt	Silt	%	Laster particle sizer method [34]
Sand	Sand	%	Laster particle sizer method [34]
pH	pH	g/kg	Soil: water suspension (1:5 *w*/*v*) by pH meter [35]
Organic matter	OM	g/kg	Potassium dichromate oxidation method [36]
Total nitrogen	TN	g/kg	The Kjeldahl method [34]
Available nitrogen	AN	mg/kg	The alkaline hydrolysis diffusion method [34]
Total phosphorus	TP	g/kg	The alkali fusion-molybdenum antimony colorimetric method [34]
Available phosphorus	AP	mg/kg	The sodium bicarbonate extraction-molybdenum antimony colorimetric method [34]
Total potassium	TK	mg/kg	The alkali fusion-flame photometer method [34]
Available potassium	AK	mg/kg	1 mol/L ammonium acetate leaching method [34]

**Table 2 ijerph-19-08235-t002:** Load matrix and Norm values for each soil indicator.

Soil Indicators	Principal Components			Norm
	PC-1	PC-2	PC-3	
BD	**−0.951**	−0.192	0.074	3.59
SWC	0.687	−0.544	−0.398	2.34
Clay	0.632	** −0.613 **	0.233	2.09
Silt	0.724	−0.051	**0.651**	2.36
Sand	−0.750	0.150	** −0.626 **	2.51
pH	−0.502	**−0.679**	0.288	1.64
OM	**0.974**	0.056	−0.027	3.72
TN	** 0.981 **	−0.006	−0.009	3.77
TP	0.575	**0.632**	−0.164	1.83
TK	−0.817	0.272	0.348	2.80
AN	**0.975**	0.137	−0.019	3.75
AP	0.649	** 0.624 **	0.113	2.16
AK	−0.660	0.602	0.368	2.27
Eigenvalue	7.832	2.578	1.414	
Variance/%	60.246	19.833	10.876	
Cumulative/%	60.246	80.079	90.955	

Note: The variable corresponding to the bold value is selective further due to its relatively high scores. Variable loading coefficients (eigenvalues) of the first three factors were extracted using 13 soil attributes, their eigenvalues, and individual and cumulative percentage of total variance explained by each factor. Factor loadings are considered highly weighted when within 10% of the variation of the absolute values of the highest factor loading in each factor. Bold-underlined soil attributes correspond to the indicators included in the MDS. See Table 1 for abbreviations.

**Table 3 ijerph-19-08235-t003:** Commonality and weight of minimum dataset and total dataset for soil quality assessment.

Soil Indicators	TDS		MDS	
	Commonality	Weight	Commonality	Weight
BD	0.946	0.080		
SWC	0.926	0.078		
Clay	0.829	0.070	0.559	0.214
Silt	0.951	0.080		
Sand	0.977	0.083	0.824	0.316
pH	0.796	0.067		
OM	0.953	0.081		
TN	0.963	0.081	0.833	0.319
TP	0.893	0.076		
TK	0.863	0.073		
AN	0.970	0.082		
AP	0.824	0.070	0.393	0.151
AK	0.934	0.079		

## Data Availability

Data sharing is not applicable.

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
