# Peer review of "Are Iron Tailings Suitable for Constructing the Soil Profile Configuration of Reclaimed Farmland? A Soil Quality Evaluation Based on Chronosequences"

_ijerph, 2022, doi:10.3390/ijerph19148235_

Round 1

Reviewer 1 Report

In this paper, the authors reported the construction of the soil profile configuration of reclaimed farmland using iron tailings. Abundant data were collected and the results were discussed in detail. I recommend it to be published after some minor revision. Detailed comments:

1.      Is there any proof verifying that the moisture was changed?

2.      Line 472. The ecological harm of the released heavy metals does not only depend on the content of heavy metals, but has close relationship with the species or the chemical states of the heavy metals. If the metal ions are easy to transport, then it should be paid more attention.

Author Response

Response to Reviewer 1 Comments

Cover letter

Thanks very much for your time to review this manuscript. We really appreciate all your comments and your help. We have considered these comments carefully and tried our best to modify and improve every one of them. With this submission, we provided a version (marked) of the revised manuscript. Responses to reviewers’ comments on the manuscript of marked are detailed below.

Detailed comments

Point 1: Is there any proof verifying that the moisture was changed?

Response 1:  Thank you very much for proposing such a good comment. Firstly, for the change of moisture, we analyzed in detail the change of soil water content (SWC) in different soil depths under different reconstruction measures and different reclamation years in Section 3.1(Line 270-276). Moreover, based on your comment, for the change of soil moisture under different topsoil covering thicknesses in the study area, the influence of different topsoil covering thicknesses on the change of reclaimed soil moisture was supplemented in the discussion section with reference to the existing similar research results and combine with the rainfall data in the study area.  Line 472-479 & Line 485-486

Point 2: Line 472. The ecological harm of the released heavy metals does not only depend on the content of heavy metals, but has close relationship with the species or the chemical states of the heavy metals. If the metal ions are easy to transport, then it should be paid more attention.

Response 2:   We want to thank you for your constructive and insightful criticism and advice on the ecological harm of the released heavy metals. For various possible harm caused by heavy metals, in addition to the heavy metal content in the reclaimed soil mentioned in the manuscript, our research group previously carried out the indoor column leaching test of reclaimed farmland with iron tailings, and the results showed that the heavy metal content in the leachate was very low and have no risk of soil and water contamination, especially the content of Cr, Pb and Cd, which was almost impossible to be detected. Therefore, based on your comments, in order to better explain farmland reconstruction with iron tailings does not exist heavy metal harm, according to the results of our previous studies, we added in Section 4.4 that heavy metals will not cause pollution in other existing states.  Line 536-539

We thank reviewer`s constructive comments, which significantly help improve our manuscript.

Reviewer 2 Report

To further improve the text, I suggest the following changes in the manuscript.
• Abstract: Abstract should be written in concise. I would suggest listing only some of the most important results to justify the implications and conclusions of the study.
• Keywords should not be included in the title. Please remove or substitute.
• The background of an introduction should be revised accordingly. This section must be upgraded with some latest references. I will suggest to read these articles and cite properly,

  • 10.3389/fenvs.2022.900193
  • 10.3389/fmats.2022.864254
  • 10.15244/pjoes/134292

https://link.springer.com/article/10.1007/s11356-021-16167-5

https://link.springer.com/article/10.1007/s11356-021-12867-0

• The introduction is very good. It doesn't reflect the goal; please rewrite it again, 
• Objectives of this study must be included at end of introduction part more clearly. 
• What is contribution of this work to existing literature? 
• It has been observed that the authors have used old references and ignored the latest studies. So it is suggested to add recent references. Please check reference section some references are missing.
• The policy implications also required elaboration. The implications should go along with the results and the course of action should be discussed in this part.
• Research Limitation and recommendations must be included in conclusion part.
• In some places, some grammatical errors are found that need to be fixed.

Reviewer 3 Report

Dear authors, 

in line 474 there is an issue with citing an article. Please check for other referencing issues.

Author Response

Response to Reviewer 3 Comments

Cover letter

Thanks very much for your time to review this manuscript. We really appreciate all your comments and your help. We have considered these comments carefully and tried our best to modify and improve them. With this submission, we provided a version (marked) of the revised manuscript. Responses to reviewers’ comments on the manuscript of marked are detailed below.

Detailed comments

Point 1:  in line 474 there is an issue with citing an article. Please check for other referencing issues.

Response 1:  Thank you very much for your careful review, so that we can correct the details in our manuscript. Based on your comments, we have revised the citation question in line 474 (Now in Line 534). Furthermore, we have carefully examined and revised the references for each citation.

We thank reviewer`s constructive comments, which significantly help improve our manuscript.

Round 2

Reviewer 2 Report

I agree with revision